# Two Cases of Vancomycin-Induced Neutropenia

**DOI:** 10.3390/pharmacy12010038

**Published:** 2024-02-19

**Authors:** Kirsten Ganaja, Sarah Scoular, Staci Hemmer

**Affiliations:** 1Dignity Health Marian Regional Medical Center, Santa Maria, CA 93454, USA; kirsten.ganaja@commonspirit.org; 2Skaggs School of Pharmacy, University of Montana, Missoula, MT 59812, USA; staci.hemmer@umontana.edu

**Keywords:** adverse drug reactions, side effects, leukopenia, white blood cell

## Abstract

(1) Background: The incidence of vancomycin-induced neutropenia in hospitalized patients is estimated to be around 2 to 8 percent Data surrounding vancomycin-induced neutropenia is limited as it is based on a small number of observational case reports. Additionally, it is difficult to provide generalized conclusions since patient characteristics and indications for treatment vary between reports. (2) Case Reports: We present two cases of vancomycin-induced neutropenia that occurred at our facility; a 50-year-old male who developed neutropenia after treatment with vancomycin for a gluteal abscess and a 51-year-old female who developed neutropenia after treatment with vancomycin for lumbar osteomyelitis. In both cases, neutropenia resolved within 2 days of discontinuation of vancomycin. (3) Conclusions: Vancomycin-induced neutropenia is thought to be a relatively uncommon adverse drug reaction. These two cases of neutropenia likely caused by prolonged exposure to vancomycin occurred at our facility within 3 months of each other. Additional studies are needed to better understand the true incidence of this adverse drug reaction and to identify risk factors that may predispose patients to vancomycin-induced neutropenia.

## 1. Background

The incidence of vancomycin-induced neutropenia in hospitalized patients is estimated to be around 2 to 8 percent [1,2]. Risk factors for general drug-associated neutropenia include age greater than 65 years old and female sex, while a risk factor specific to vancomycin-induced neutropenia is prolonged drug exposure [2,3]. Current evidence shows no relationship between the development of neutropenia and total daily dose or serum concentration of vancomycin [1]. In addition to vancomycin, medications which can cause drug-induced neutropenia include beta-lactam antibiotics, diclofenac, spironolactone, carbamazepine, sulfamethoxazole-trimethoprim, and antithyroid medications such as propylthiouracil [3]. Neutropenia often occurs after at least 12 days of vancomycin therapy, but has been observed as early as 7 days after treatment initiation [4,5]. A more rapid onset may occur if patients with a history of vancomycin-induced neutropenia are re-exposed to vancomycin [4]. Treatment involves immediate discontinuation of vancomycin, as neutrophils will spontaneously recover once the offending agent is removed [4]. Neutropenia typically resolves within 48 to 72 h, although time to resolution may be increased in patients receiving hemodialysis [5]. The granulocyte colony-stimulating factor (G-CSF) filgrastim has also been used as adjunct therapy in some case reports of severe drug-induced agranulocytosis as it can accelerate neutrophil recovery time [3,4,5,6].

Data surrounding vancomycin-induced neutropenia is limited as it is based on a small number of observational case reports. Additionally, it is difficult to provide generalized conclusions since patient characteristics and indications for treatment vary between reports. Case reports published within the past 10 years from reputable sources are summarized in Table 1. In this case report, we describe two incidences of neutropenia that occurred after prolonged exposure to vancomycin. The first case is of a 50-year-old male who was transferred to our facility for acute treatment of a gluteal abscess. The second case is of a 51-year-old female who was admitted for inpatient rehabilitation for lumbar spinal stenosis with lumbar myelopathy.

## 2. Case Reports

### 2.1. Patient Case 1

A 50-year-old white male was transferred to our facility for treatment of a right gluteal abscess secondary to blunt force trauma. The patient’s pertinent medical history included uncontrolled type 2 diabetes mellitus, binge drinking, tobacco use, and former intravenous drug use. He had no known drug allergies. At the outside facility, partial drainage was performed, samples were sent for culture, and the patient was given 6 days of ceftriaxone and vancomycin before being transferred to our facility. On admission to our emergency department, purulent fluid was visible from the previous incision. Incision and drainage were performed and 4 drains were placed. Cultures from the outside facility obtained from the partial drainage of the abscess grew methicillin-susceptible *Staphylococcus aureus* (MSSA), resistant to penicillin and ampicillin and sensitive to all other agents including levofloxacin. Due to purulence and worsening patient symptoms and vitals, the infectious disease team recommended continuing treatment with vancomycin and ceftriaxone rather than de-escalating. Concomitant medication during his hospitalization included, furosemide, insulin lispro, insulin glargine, chlordiazepoxide, morphine, oxycodone, acetaminophen, gabapentin, midazolam, ondansetron, lactated ringers, fluoxetine, potassium chloride, magnesium sulfate, nicotine patch, heparin and lisinopril. On admission, the patient weighed 107.2 kg and laboratory results showed a white blood cell (WBC) count of 6000 cells/mcL with 73.3% neutrophils, serum creatinine 0.57 mg/dL, glucose 193 mg/dL, and hemoglobin A1c (HbA1c) 12.9%. The calculated absolute neutrophil count (ANC) was 4398 cells/mcL. A computed tomography (CT) scan of the pelvis revealed a right gluteal abscess with gas and fluid as well as scrotal edema. The patient received a loading dose of vancomycin 2 g IV one time and ceftriaxone 1 g IV every 24 h. A vancomycin trough concentration was measured within the first day of therapy, 22 h after the first dose via blood sample. A maintenance dose of 1.5 g IV every 8 h was selected to achieve a predicted vancomycin area-under-the-curve (AUC) of 550 mg·h/L and a trough of 15 mg/L. This dose was selected based on Bayesian assessment of vancomycin level to maintain AUC between 400–600 mg·h/L. A testicular ultrasound was performed which showed no evidence of fluid collection, abscess, or Fournier’s gangrene.

Eleven days after initiation of vancomycin, magnetic resonance imaging (MRI) showed no signs of septic arthritis of the hip joint or osteomyelitis but did show signs of continued yet improving infection and gluteal myositis. After pharmacokinetic analysis, the vancomycin treatment regimen was adjusted to 1.25 g IV every 8 h to achieve an AUC of 513 mg·h/L and a trough of 14.9 mg/L. On day 13 of therapy, WBC count was 4800 cells/mcL with 82.8% neutrophils and a calculated ANC of 3974 cells/mcL. The patient had also developed a rash which was thought to be due to ceftriaxone and the medication was changed to levofloxacin. On day 15 of vancomycin therapy, the nadir was reached with a WBC count of 2400 cells/mcL, 53.1% neutrophils and an ANC of 1274 cells/mcL. This patient had no additional adverse drug events. A repeat CT scan of the pelvis revealed no undrained fluid collections and vancomycin was discontinued. One day after discontinuation of vancomycin, on hospital day 16, the patient spiked a fever of 38.9 and became tachycardic, meeting two out of four systemic inflammatory response (SIRS) criteria, which required intervention by a rapid response team. SIRS was defined as having two or more of the following: temperature greater than 38 °C or less than 36 °C, heart rate greater than 90 beats per minute, respiratory rate greater than 20 breaths per minute, and WBC count greater than 12,000 cells/mm^3^ or less than 4000 cells/mm [3,7]. Urine and blood cultures were ordered and a chest X-ray was performed which revealed bilateral lower lobe pneumonia. Meropenem was initiated and levofloxacin was continued for empiric coverage of hospital acquired pneumonia. On day 16, after discontinuation of vancomycin, the patient’s WBC count was 3000 cells/mcL with 43.3% neutrophils and an ANC of 1299 cells/mcL. By day 17, the patient’s temperature and vital signs were within normal limits, although the patient had continued altered mental status. MRI results were unremarkable, and the patient was diagnosed with metabolic encephalopathy. The patient’s WBC count continued to increase to 4100 cells/mcL with 39.3% neutrophils and an ANC of 1611 cells/mcL. By day 20, the patient was alert and oriented and his WBC count was 5300 cells/mcL. He continued to stabilize and was discharged on oral levofloxacin two days later. Co-administered medications were analyzed for drug-drug interactions and adverse reactions and were found to not be contributing to neutropenia. The adverse drug reaction (ADR) probability was calculated as a score of 4 using the Naranjo Algorithm. The Naranjo algorithm was developed in 1991 as a way to assess causality for adverse drug reactions. Scores range from −4 to 13 [8]. The score indicates that vancomycin was a “possible” cause since there are previous conclusive reports of this ADR, neutropenia occurred after vancomycin was administered, the ANC improved after vancomycin was discontinued, and the ADR was confirmed by WBC and neutrophil laboratory results [8]. However, the patient received ceftriaxone for 13 days before neutropenia developed, and could have been developing sepsis which could both be a possible alternative cause for neutropenia. Figure 1 summarizes the hospital stay.

### 2.2. Patient Case 2

A 51-year-old white female was transferred to our facility with bilateral lower extremity edema, weakness, and neuropathic pain, and was admitted for comprehensive inpatient rehabilitation for lumbar spinal stenosis with lumbar myelopathy. The patient’s pertinent medical history included prior methicillin-resistant *Staphylococcus aureus* (MRSA) infection, osteomyelitis of the spine, spondylolisthesis of the lumbar region, tobacco use, and infrequent alcohol use. Drug allergies included codeine, paroxetine, and sulfanilamide. Concomitant medication during his hospitalization included, docusate, duloxetine, enoxaparin, gabapentin, oxycodone, senna, acetaminophen, methocarbamol, melatonin and ondansetron. Physical examination of the patient’s surgical incision on the lumbar spine revealed a moderate amount of scabbing, no drainage, and no signs of infection. On admission, the patient weighed 101 kg and laboratory results showed WBC count 6400 cells/mcL with 70.7% neutrophils, and serum creatinine 0.72 mg/dL. The calculated absolute neutrophil count (ANC) was 4525 cells/mcL. At the outside facility, she was started on vancomycin 1.25 g IV every 12 h for lumbar osteomyelitis, which was continued at our facility upon transfer. She did not receive a loading dose. The duration of antibiotic therapy was estimated to be 6–12 weeks, as determined by an infectious disease consult.

After pharmacokinetic analysis on day 3 of hospitalization, the vancomycin treatment regimen was decreased to 1 g IV every 12 h to which gave a predicted trough of 16.1 mg/L and an AUC of 579 mg·h/L. On day 7, the patient’s WBC count began trending downward. Her WBC count was 3400 cells/mcL with 50.9% neutrophils and an ANC of 1731 cells/mcL. At this point, more frequent WBC monitoring was ordered due to concern for vancomycin-induced neutropenia. By day 9, the patient’s WBC count decreased to 2600 cells/mcL with 46.3% neutrophils and an ANC of 1204 cells/mcL. An infectious disease consult was then requested, and the decision was made to discontinue vancomycin and start daptomycin 800 mg IV every 24 h (8 mg/kg). The WBC count increased to 3300 cells/mcL with 53% neutrophils and an ANC of 1749 cells/mcL two days after discontinuing vancomycin therapy.

Neutropenia was observed on day 14 with a WBC count of 2500 cells/mcL, 40.9% neutrophils and an ANC of 1023 cells/mcL. This was observed after no changes were made to the patient’s antibiotic therapy. By day 18, the WBC count increased to 4400 cells/mcL with 43% neutrophils and an ANC of 1892 cells/mcL, then returned to a WBC count of 2500 cells/mcL with 41.8% neutrophils and an ANC of 1045 cells/mcL on day 21. The patient remained afebrile, had no other signs or symptoms of infection, and no changes were made to antimicrobial therapy. On day 23, the WBC count was 3300 cells/mcL with 52.5% neutrophils and an ANC of 1733 cells/mcL, and then decreased again to 2900 cells/mcL with 46.5% neutrophils and an ANC of 1349 cells/mcL the following day. This patient had no additional adverse drug events.

The patient was discharged on day 25 with outpatient physical therapy and home health nursing for antibiotic infusions. She was continued on daptomycin 800 mg IV every 24 h with a plan to monitor WBC count weekly. These outpatient lab values were not available. Co-administered medications were analyzed for drug-drug interactions and adverse reactions and were found to not be contributing to neutropenia. The adverse drug reaction probability was calculated as a score of 6 using the Naranjo Algorithm [8]. This score indicates that vancomycin was a “probable” cause due to previous conclusive reports of this ADR, neutropenia occurring after administration of vancomycin, no alternative causes of neutropenia, and confirmation of the ADR by WBC and neutrophil laboratory results [8]. A summary of the patient’s inpatient rehabilitation stay is illustrated in Figure 2.

## 3. Discussion

Neutrophils are a critical component of the innate immune system, and drug-induced neutropenia places patients at increased risk of severe infections, septicemia, and septic shock [3]. Neutropenia is generally defined as an ANC less than 1500 cells/mcL and severe neutropenia, known as agranulocytosis, is considered to be an ANC less than 500 cells/mcL [3]. Vancomycin-induced neutropenia has often been defined as an ANC less than 1000 cells/mcL [2]. The mechanism for drug-induced immune neutropenia (DIIN) is not fully understood, and a variety of hypotheses have been proposed [3].

Literature and previous case reports suggest closely monitoring patients receiving vancomycin for longer than seven days [1,5,6,9]. Both of our patients had serious infections requiring prolonged vancomycin exposure, placing them at high risk for vancomycin-induced neutropenia. Decreased renal clearance is also considered to be a risk factor for this adverse drug reaction; however, both of our patients had adequate renal function [6]. As seen in the case report by Duff et al., rechallenging patients with a history of vancomycin-induced neutropenia is not encouraged due to limited studies surrounding safety of this practice and the risk of rapid development of agranulocytosis [6]. Vancomycin was not restarted in either of our patients, and alternative antibiotics were administered. Some case reports have used granulocyte colony-stimulating factor (G-CSF) to accelerate neutrophil recovery time [3]. In our two cases, G-CSF therapy was unnecessary as the neutropenia was mild, vancomycin was discontinued at the onset of neutropenia and neutrophils recovered after vancomycin discontinuation.

In case 1, a decrease in WBC count was first seen on Day 10 of vancomycin treatment and neutropenia was observed on Day 15, when vancomycin was discontinued. The WBC count and ANC were within normal limits one day after discontinuing therapy. WBCs continued to increase slightly over the next four days. Although this was a medically-complex patient, the time to neutropenia onset and recovery is similar to that of previously published case reports. It is important to note that beta-lactam antibiotics have been shown to cause drug-induced neutropenia, and this patient received fourteen days of concomitant ceftriaxone.

In case 2, a decrease in WBC count was first seen on Day 7 of vancomycin treatment, and neutropenia first resolved within 2 days of discontinuing vancomycin therapy. However, the patient’s WBC count continued to oscillate after switching to daptomycin and neutropenia occurred on Day 14 and again on Day 21 of her hospital stay. The time to complete WBC count recovery was unknown, as the neutropenia returned the day prior to discharge and subsequent laboratory results were obtained at an outpatient facility. It is important to note that laboratory data was not regularly obtained during the first week of the patient’s hospital stay, so it is unknown if the decrease in WBC count may have occurred earlier than on Day 7 of vancomycin therapy. Although the time to neutropenia onset was similar to that of previous case reports, the oscillating pattern of the WBC count after switching from vancomycin to daptomycin was not similar to that of previous publications. However, ANC recovery times greater than 14 days have been reported [2,9].

## 4. Conclusions

Vancomycin-induced neutropenia is thought to be a relatively uncommon adverse drug reaction. However, these two cases of neutropenia likely caused by prolonged exposure to vancomycin occurred at our facility within 3 months of each other. Additional studies are needed to better understand the true incidence of this adverse drug reaction and to identify risk factors that may predispose patients to vancomycin-induced neutropenia.

## Figures and Tables

**Figure 1 pharmacy-12-00038-f001:**
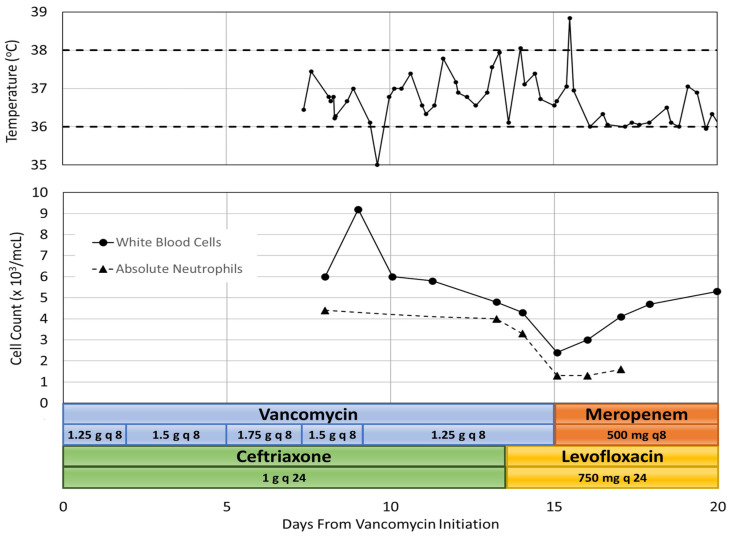
Case 1 Summary of Temperature, Cell Count and Antibiotic Course.

**Figure 2 pharmacy-12-00038-f002:**
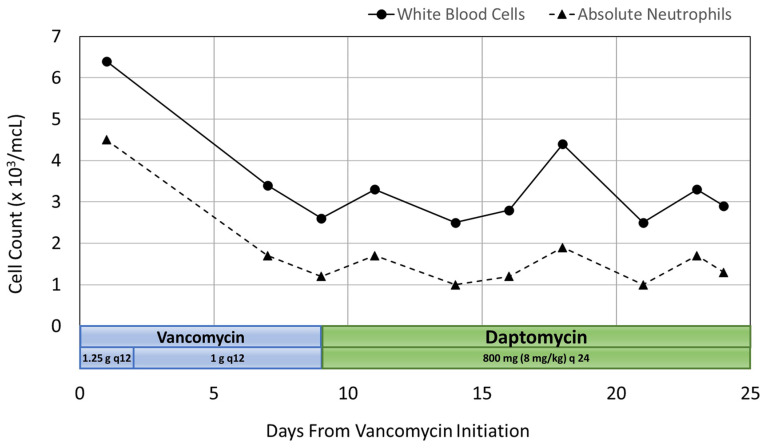
Case 2 Summary of Cell Count and Antibiotic Course.

**Table 1 pharmacy-12-00038-t001:** Vancomycin-induced neutropenia: summary of case reports.

Reference	Patient Age (Years), Sex	Vancomycin Indication (s)	LNC (Cells/mcL)	Time to LNC (Days)	Concurrent Medications	Intervention	Time to Neutrophil Recovery
Di Fonzo et al. [1] (2018)	38, M	Kidney abscess, Infective Endocarditis	60	31	heparin omeprazole insulin paracetamol	Switched vancomycin to daptomycin. 60 MU of G-CSF	2 days
Lintel et al. [4] (2021)	46, F	Osteomyelitis with epidural phlegmon	900	33	ceftriaxone cefepime metronidazole acetaminophen oxycodone cyclobenzaprine pregabalin	Switched vancomycin to daptomycin	5 days
Shaukat et al. [5] (2017)	19, M	Axillary and perianal abscesses, infective arthritis of sternoclavicular joint	700	14	ceftriaxone	Switched vancomycin to clindamycin + rifampicin	3 days
Duff et al. [6] (2012)	78, F	Superficial abdominal wall abscess, septic thrombophlebitis	600	Initial: 56 Retrial: 4	aztreonam amikacin prednisone lisinopril aspirin simvastatin glipizide	Initial: filgrastim 300–480 mcg daily, methylprednisolone 1 mg/kg daily Retrial: Discontinue vancomycin, cyclosporine 150 mg BID for 2 months	3 weeks

Abbreviations: LNC, lowest neutrophil count; Time to LNC, days since vancomycin initiation.

## Data Availability

The original contributions presented in the study are included in the article, further inquiries can be directed to the corresponding author.

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
