# Peer review of "Two Cases of Vancomycin-Induced Neutropenia"

_pharmacy, 2024, doi:10.3390/pharmacy12010038_

Round 1

Reviewer 1 Report

Comments and Suggestions for Authors

The case reports presents two cases of vancomycin-induced neutropenia. The authors included in the title also a ‘Summary of Reported Literature’, however, in my opinion, the manuscript does not include this summary. Moreover, it is not discussed in the Discussion as well as the number of cited articles (9) suggests that it is rather not a summary. I suggest deleting this part of the title, especially as it was submitted as a case report, which basically, is thought to be short.

My other remarks are as follows:

·         The time of the observation after discontinuation of vancomycin is different in both cases. Moreover, the changes in the cell count is more variable in the female patient with the same low lower on day 21 as it was on day 9 when the vancomycin was withdrawn.

·         Both patients were treated with numerous medicines. Were all the possible interactions taken into account? Does any of the co-administered drugs cause neutropenia? If the co-administered drugs were carefully analyzed and considered to be free of neutropenia adverse effect it should be clearly stated in the manuscript.

·         In Discussion, the authors mentioned ceftriaxone, which is known to induce neutropenia and was administered to of the patients. However the author referred to Table 2, which was not included in the manuscript, therefore it is difficult to understand this result. If the authors intended to refer to Fig. 1, the time interval between ceftriaxone discontinuation and vancomycin discontinuation is only one day, which is, in my opinion, too short to draw the conclusion that ‘vancomycin was more likely to be the offending agent’ rather than ceftriaxone.

·         The conclusions in the abstract are far-fetched based on the results presented in the manuscript. Moreover, these conclusions are not included, neither discussed in the manuscript.

Author Response

We greatly appreciate the thoughtful comments from Reviewer 1. Responses are shown after each comment in bold

The case reports presents two cases of vancomycin-induced neutropenia. The authors included in the title also a ‘Summary of Reported Literature’, however, in my opinion, the manuscript does not include this summary. Moreover, it is not discussed in the Discussion as well as the number of cited articles (9) suggests that it is rather not a summary. I suggest deleting this part of the title, especially as it was submitted as a case report, which basically, is thought to be short. Agree. Summary of reported literature is now deleted from the title.

My other remarks are as follows:

  • The time of the observation after discontinuation of vancomycin is different in both cases. Moreover, the changes in the cell count is more variable in the female patient with the same low lower on day 21 as it was on day 9 when the vancomycin was withdrawn. Yes we agree with this statement. We have a statement regarding this in the discussion regarding that recovery of neutrophils can take up to 14 days.
  • Both patients were treated with numerous medicines. Were all the possible interactions taken into account? Does any of the co-administered drugs cause neutropenia? If the co-administered drugs were carefully analyzed and considered to be free of neutropenia adverse effect it should be clearly stated in the manuscript. All co-administered medications were analyzed. We have added in statements to let the reader know this in both patient cases. Thank you!
  • In Discussion, the authors mentioned ceftriaxone, which is known to induce neutropenia and was administered to of the patients. However the author referred to Table 2, which was not included in the manuscript, therefore it is difficult to understand this result. If the authors intended to refer to Fig. 1, the time interval between ceftriaxone discontinuation and vancomycin discontinuation is only one day, which is, in my opinion, too short to draw the conclusion that ‘vancomycin was more likely to be the offending agent’ rather than ceftriaxone. Agree. This comment was removed. 
  • The conclusions in the abstract are far-fetched based on the results presented in the manuscript. Moreover, these conclusions are not included, neither discussed in the manuscript. Agree. Conclusion in abstract was changed to better reflect the results presented in the manuscript.

Reviewer 2 Report

Comments and Suggestions for Authors

The manuscript presents two case reports of vancomycin-induced neutropenia, which is an important topic.

1.       Here are some comments and questions regarding the manuscript:

2.       Please clarify the blood sample timing of blood sample collection for trough concentration and the method for vancomycin concentration measurement.

3.       In line 139, it is mentioned that the predicted area under the curve (AUC) and trough concentration values are 579 mg*h/L and 16.1 mg/L, respectively. It would be helpful to clarify whether these values were predicted or measured in the study.

4.       It would be valuable to include the percentage of neutropenia cases observed in the facility over the study period. This information helps verify the relative frequency or incidence of vancomycin-induced neutropenia.

5.       It would be informative to know if any additional ADEs were observed in the two reported cases of vancomycin-induced neutropenia.

6.       Typo correction: In line 85, "1.25 mg" should be corrected to "1.25 g".

Comments on the Quality of English Language

N/A

Author Response

We greatly appreciate the thoughtful comments from Reviewer 2. Responses are shown after each comment in bold

  1. Please clarify the blood sample timing of blood sample collection for trough concentration and the method for vancomycin concentration measurement. Thank you for this comment. Details including timing and method for vancomycin concentrations were added. 
  2. In line 139, it is mentioned that the predicted area under the curve (AUC) and trough concentration values are 579 mg*h/L and 16.1 mg/L, respectively. It would be helpful to clarify whether these values were predicted or measured in the study. "Predicted" was added to clarify this statement.
  3. It would be valuable to include the percentage of neutropenia cases observed in the facility over the study period. This information helps verify the relative frequency or incidence of vancomycin-induced neutropenia. Thank you. Unfortunately we do not have the time to obtain this information prior to the requested re-submission.
  4. It would be informative to know if any additional ADEs were observed in the two reported cases of vancomycin-induced neutropenia. The additional ADE of rash was discussed in patient case 1. A statement was added to both cases to let readers know there were no additional ADEs.
  5. Typo correction: In line 85, "1.25 mg" should be corrected to "1.25 g". Corrected. Thank you!